# Dynamic Changes in Serum Metabolomic Profiles of Growing Pigs Induced by Intravenous Infusion of Sodium Butyrate

**DOI:** 10.3390/metabo10010020

**Published:** 2020-01-01

**Authors:** Hongyu Wang, Erdu Ren, Xiaoe Xiang, Yong Su, Weiyun Zhu

**Affiliations:** 1Laboratory of Gastrointestinal Microbiology, Jiangsu Key Laboratory of Gastrointestinal Nutrition and Animal Health, College of Animal Science and Technology, Nanjing Agricultural University, Nanjing 210095, China; 2019205029@njau.edu.cn (H.W.); 2015105044@njau.edu.cn (E.R.); zhuweiyun@njau.edu.cn (W.Z.); 2National Center for International Research on Animal Gut Nutrition, Nanjing Agricultural University, Nanjing 210095, China; 3National Experimental Teaching Demonstration Center of Animal Science, Nanjing Agricultural University, Nanjing 210095, China; xxxiang@njau.edu.cn

**Keywords:** sodium butyrate, serum metabolomic, intravenous infusion, growing pig

## Abstract

This study aimed to explore the dynamic changes in metabolite profiles and metabolism pathways in the serum of growing pigs by intravenous infusion of sodium butyrate (SB). Fourteen crossbred growing barrows (BW = 23.70 ± 1.29 kg) fitted with jugular cannula were randomly allocated to the SB and control (Con) groups, each group consisted of seven replicates (pens), with one pig per pen. At 9:00 of each day during the experimental period, pigs in the SB group were infused with 10 mL of SB (200 mmol/L, pH 7.4, 37 °C) via precaval vein, while the Con group was treated with the same volume of physiological saline. On day 4, the blood of each pig was collected at 0, 30, 60, and 120 min after the intravenous infusion. Metabolites in the serum were detected by gas chromatograph-mass spectrometry analysis. Pathway analysis of metabolomic profiles showed that the differential metabolites mainly enriched in amino acid metabolism, lipid-related metabolism, and the tricarboxylic acid (TCA) cycle. More importantly, the relative concentrations of all eight essential amino acids, five non-essential amino acids, and two amino acid derivatives were decreased by the parenteral SB. In addition, SB significantly increased the relative concentrations of eicosanoic acid and octadecanoic acid and decreased the relative concentration of glycerol-3-phosphate at 0 min (three days after intravenous infusion of SB), which suggests that parenteral SB may increase stearates mobilization and decrease the biosynthesis of stearates. In conclusion, intravenous infusion of SB may induce more amino acids to synthesize proteins and affect fat metabolism through increasing fat mobilization and decreasing the biosynthesis of stearates. However, a further study is needed to understand the mechanism of extensive metabolic pathway changes induced by parenteral SB.

## 1. Introduction

Butyrate is one of the short-chain fatty acids (SCFA) with four-carbon, which is mainly produced by microbial fermentation [1,2]. It has received particular attention for its beneficial effects on improving immunity and intestinal health [3,4], regulating intestinal microecology [5,6], inhibiting bacterial growth [7], and promoting growth [8]. Therefore, butyrate preparations (such as sodium butyrate (SB) and butyrate glycerides) have been widely used in pig [9,10], poultry [11,12] and aquatic [13,14] feed. It is well documented that butyrate is the major energy source for colonocytes [15]. As regulating molecule, butyrate exerts an effect on energy metabolism based on G-protein coupled receptors (GPCR) [16], and butyrate is a typical histone deacetylase (HDAC) inhibitor which mediates physiological function by inducing histone hyperacetylation to improve genetic transcription [17]. Meanwhile, it was proven that butyrate could increase the muscle fiber cross-sectional area, prevent intramuscular fat accumulation, and improve glucose metabolism in aging mice [18]. Overall, butyrate plays an important role in maintaining nutrient metabolism and the health of an animal.

Interestingly, a study has shown that 90% of the butyrate produced in the gut was used by colon cells [19], with only a small portion going into the bloodstream. Even so, butyrate still plays a big role in body metabolim. Several reports in vitro showed that butyrate could stimulate porcine adipocyte differentiation [20], induce metabolic adaptations [21], and stimulate leptin production in adipocytes [22]. Moreover, researches have found that intravenous injection of SB could increase the secretion of amylase [23], improve structural aspects of intestinal adaptations by increasing proliferation and decreasing apoptosis [24], change the gene expression of the intestine [25] and the liver [26], and remold the composition of intestinal microecology [27] in animal models with intestine injury. Our previous study [26] showed that the infusion of butyrate changed the metabolite profile of the liver in a healthy growing pig model. However, limited information is available on the impact of parenteral butyrate on the profile of serum metabolites and its dynamics changes in pigs.

In this study, we hypothesized that the parenteral butyrate may have extensive effects on amino acid metabolism, fat metabolism, and energy metabolism accompanied with the dynamic changes of metabolite concentrations participated in these pathways. Therefore, the purpose of this study was to explore the dynamic change of the metabolite profile and metabolism pathway in the serum of growing pigs by intravenous butyrate through gas chromatograph-mass spectrometry (GC-MS) analysis.

## 2. Results

### 2.1. Effects of Intravenous Infusion of SB on Serum Biochemical Parameters

Serum metabolite concentrations of pigs in control (Con) and SB groups are shown in Table 1. The infusion of SB increased the concentration of high-density lipoprotein-cholesterol (HDL-C) (*p* = 0.005), while there was no difference in the concentrations of low-density lipoprotein-cholesterol (LDL-C), glucose, and triglyceride (*p* > 0.05) between two groups (Table 1). The main effect of time and interaction effect was not detected between groups at 0, 30, 60, and 120 min.

### 2.2. Global Overview of the Serum Metabolomic Profile

The standard deviations of internal standards in Quality Control (QC) were both below 1% (Appendix A). With the GC-MS method, 232 peaks were obtained. A total of 75 metabolites including fatty acid, amino acid, organic acid, saccharides, alcohols, amine, pyrimidone, and purines (Appendix A) were identified in the serum of pigs in both Con and SB groups.

Time-series analysis combined with one experimental factor method was used to explore the main effects of time and SB infusion on the serum metabolites and their interaction effect. These results (Appendix A) indicated that 48 metabolites were affected by the infusion of SB (*p* < 0.05) and 19 metabolites were changed with time (*p* < 0.05). Among them, only 11 metabolites had an interaction effect (*p* < 0.05). Pathway enrichment analysis showed that the influenced metabolism pathway mainly enriched in alanine, aspartate and glutamate metabolism, arginine and proline metabolism, glycine, serine and threonine metabolism, butanoate metabolism, glycerophospholipid metabolism, and the tricarboxylic acid (TCA) cycle (Figure 1).

Venn digrams of differential metabolites and enriched metabolic pathways at different timepoints showed that two differential metabolites (methionine and tyrosine) and one metabolic pathway (phenylalanine tyrosine and tryptophan biosynthesis) were all influenced at four timepoints (Figure 2).

### 2.3. Effects of Intravenous Infusion of SB on Serum Metabolomics of Growing Pigs

As shown in Figure 3 and Figure 4, the score plots of both principal component analysis (PCA) and partial least squares discriminant analysis (PLS-DA) distinguished the SB and Con groups at 0, 30, 60, and 120 min after SB infusion.

At 0 min, just before intravenous infusion of day 4 and actually three days after the first intravenous infusion, 20 metabolites were identified through combination methods of univariate and multivariate analysis (Appendix A). Pathway enriched analysis results (Figure 5, T0) indicated that these 20 differential metabolites mainly enriched in alanine, aspartate and glutamate metabolism, D-glutamine and D-glutamate metabolism, phenylalanine metabolism, arginine biosynthesis, phenylalanine, tyrosine and tryptophan biosynthesis, and TCA cycle.

At 30 min, 17 differential metabolites were identified compared with the Con group (Appendix A). These differential metabolites mainly enriched in alanine, aspartate and glutamate metabolism, arginine and proline metabolism, beta-alanine metabolism, glycine, serine and threonine metabolism, phenylalanine metabolism, D-Glutamine and D-glutamate metabolism, arginine biosynthesis, phenylalanine, tyrosine and tryptophan biosynthesis, and glycerolipid metabolism (Figure 5, T1).

At 60 min, seven differential metabolites were identified respectively of the SB group compared with the Con group (Appendix A). These differential metabolites mainly enriched in phenylalanine, tyrosine and tryptophan biosynthesis, cysteine and methionine metabolism, tryptophan metabolism, and tyrosine metabolism at 60 min (Figure 5, T2).

At 120 min, 28 differential metabolites were identified of the SB group compared with the Con group (Appendix A). These differential metabolites mainly enriched in alanine, aspartate and glutamate metabolism, phenylalanine, tyrosine and tryptophan biosynthesis, cysteine and methionine metabolism, the TCA cycle, and glyoxylate and dicarboxylate metabolism (Figure 5, T3).

### 2.4. Effects of Intravenous Infusion of SB on the Dynamic Changes of Differential Metabolites

The dynamic changes of relative concentrations of mainly differential metabolites are shown in Figure 6, Figure 7 and Figure 8. In the current study, seventeen of the 20 amino acids (except histidine, arginine, and aspartic acid) were detected in the serum of growing pigs from both SB and Con groups (Appendix A). The intravenous infusion of SB significantly decreased the relative concentrations of all eight essential amino acid detected in the serum and included lysine (30, 60, and 120 min, *p* < 0.05), methionine (0, 30, 60, and 120 min, *p* < 0.05, 0.01, 0.01, and 0.05, respectively), phenylalanine (0 and 30 min, *p* < 0.05), leucine (0, 30, and 60 min, *p* < 0.05, 0.01, and 0.01, respectively), isoleucine (0, 30, and 60 min, *p* < 0.01, 0.001, and 0.001, respectively), threonine (0, 30, 60, and 120 min, *p* < 0.05, 0.01, 0.01, and 0.05, respectively), valine (30, 60, and 120 min, *p* < 0.01), and tryptophan (0 and 30 min, *p* < 0.05), five non-essential amino acids, proline (0, 30, and 60 min, *p* < 0.05, 0.01, and 0.01, respectively), serine (0, 30, and 60 min, *p* < 0.05, 0.01, and 0.05, respectively), tyrosine (0, 30, 60, and 120 min, *p* < 0.05), cysteine (30, 60, and 120 min, *p* < 0.001, 0.001, and 0.01, respectively), and glutamine (0, 30, and 120 min, *p* < 0.01, 0.05, and 0.05, respectively), and two amino acid derivatives, β-alanine (0, 30, and 120 min, *p* < 0.01, 0.05, and 0.05, respectively) and ornithine (0, 30, 60, and 120 min, *p* < 0.01, 0.01, 0.05, and 0.05, respectively).

The intravenous infusion of SB significantly increased the relative concentrations of eicosanoic acid (*p* < 0.001) and octadecanoic acid (*p* < 0.001) at 0 min while no difference was detected at 30, 60, and 120 min (*p* > 0.05). Meanwhile, the infusion of SB significantly decreased the relative concentrations of arachidonic acid at 0, 30, and 60 min (*p* < 0.05) and nonanoic acid at 30 min (*p* < 0.05). Moreover, the relative concentrations of glycerol-3-phosphate (0 min, *p* < 0.01) and glyceric acid (0, 30, and 60 min, *p* < 0.05, 0.01, and 0.05, respectively) were significantly decreased. Moreover, the relative concentrations of α-ketoglutaric acid (0 min, *p* < 0.001), citric acid (0, 60, and 120 min, *p* < 0.01, 0.05, and 0.05, respectively), and oxalic acid (0, 30, and 120 min, *p* < 0.001, 0.05, and 0.05, respectively) were significantly decreased by the infusion of SB.

## 3. Discussion

The effects of SB on the health and metabolism of pigs have been widely confirmed [28,29,30,31]. Likewise, the results in this experiment proved that intravenous infusion of SB had comparatively extensive effects on the metabolism of pigs, mainly focusing on amino acid metabolism, fatty acid metabolism, and the TCA cycle, especially on amino acid metabolism.

### 3.1. Effects of Intravenous Infusion of SB on Amino Acid Metabolism of Growing Pigs

While intensive studies mainly focused on the impact of butyrate on glycolipid metabolism, little is known on amino acid metabolism. In this study, it is surprising that the parenteral SB had a profound effect on amino acid metabolism. These metabolites were mainly enriched in alanine, aspartate and glutamate metabolism, phenylalanine, tyrosine and tryptophan biosynthesis, cysteine and methionine metabolism, phenylalanine metabolism, D-Glutamine and D-glutamate metabolism, and arginine biosynthesis. Consistently, Yu et al. (2017) [32] reported that oval butyrate changed beta-alanine metabolism, alanine metabolism, and cysteine metabolism in d7 and glycine, serine and threonine metabolism in d 21 in the liver of neonatal piglets. In addition, previous research proved that intravenous infusion of SB could change the pathway of glycine, serine and threonine metabolism, beta-Alanine metabolism, and methionine metabolism in the liver of growing piglets [26]. These results suggested that whether oral or intravenous infusion, SB could affect amino acid metabolism of pigs (newborn or growing pigs). The study herein demonstrated a more extensive role of parenteral SB in amino acid metabolism. Surprisingly, the relative concentrations of all detected differential metabolites including amino acids, fatty acids, and carbonhydrates were decreased after intravenous infusion of SB. Thus, we may speculate that parenteral SB enhanced the metabolism of nutrients (especially amino acids) in the body two hours after infusion. However, it is not clear whether this impact can last all day long. In addition, as a short-term experiment, no difference in the growth performance was observed between two groups. In the current study, there is insufficient knowledge to explain the molecular mechanism of the effects of parenteral SB on amino acid metabolism.

### 3.2. Effects of Intravenous Infusion of SB on Fat Metabolism of Growing Pigs

HDL-C is synthesized by the liver and can transport cholesterol from the extrahepatic tissues to the liver for metabolism. It has a negative correlation with the incidence of coronary heart disease, atherosclerosis, and other diseases [33]. In accordance with the research results of Jian et al. (2016) in mice [34], the results herein suggested that the infusion of SB increased serum HDL-C concentration. However, some previous studies showed that butyrate had no effect or reduced HDL-C concentration in serum. The reason for the inconsistent results may be the difference in sampling time and experimental animals, even a difference in the dose of SB [35,36,37]. As a crucial substance in fat metabolism, HDL-C is highly correlated to lipid metabolism pathway [38]. Specifically, our results showed that parenteral SB increased the relative concentration of long-chain saturated fatty acids eicosanoic and octadecanoic acid and decreased the relative concentration of glycerol-3-phosphate, which indicates that adipokinetic action is increased and fat synthesis is decreased by intravenous infusion of SB. Previous studies also proved that butyrate alleviated high fat diet-induced fat accumulation [34] and affected lipid metabolism in many ways [39,40,41]. In addition, pathway enrichment analysis results showed that arachidonic acid metabolism, glyoxylate and dicarboxylate metabolism, and glycerophospholipid metabolism were affected by the intravenous infusion of SB. In line with the result of this study, our previous study also showed that lipid metabolism-related genes and liver lipid metabolism profiles were regulated by the infusion of SB [26].

### 3.3. Effects of Intravenous Infusion of SB on the TCA Cycle of Growing Pigs

TCA cycle is the most effective way for animals to obtain energy, and it is the hub of the metabolism of sugars, lipids, and amino acids [42,43]. In line with the results of Yu et al. (2016) [32], the intervention of SB could affect the TCA cycle. The relative concentrations of key substances in the TCA cycle of citric acid, a-ketoglutaric acid, and oxalic acid were all decreased, which may signify that parenteral SB has changed the metabolism of TCA cycle. However, these effects were not consistent at different timepoints. Since the molecular mechanism of the effect of parenteral SB on TCA is still unclear, we deduced that butyrate may exert its effects on the TCA cycle in the following ways. Firstly, as free fatty acid butyrate can be oxidized through β-oxidation pathway in which a fatty acid is broken down to an acetyl-CoA molecule via a number of intermediate steps [44]. Acetyl-CoA can entirely oxidize into CO2 to generate ATP or further participate in other metabolic pathways, such as long-chain fatty acid biosynthesis, cholesterol synthesis, and ketone body formation. Secondly, SB can indirectly affect the TCA cycle by affecting fatty acid metabolism and amino acid metabolism, which was firmly confirmed in this study. Thirdly, studies showed that butyrate can influence energy metabolism through stimulating mitochondrial function [34] and affecting GPCR [16].

## 4. Material and Methods

### 4.1. Ethics Statement

This study was approved by the Nanjing Agricultural University Animal Care and Use Committee (Nanjing, Jiangsu Province, China) (SYXK2017-0027). All surgical and animal care procedures in the experiment were operated according to the standard of Experimental Animal Care and Use Guidelines of China (EACUGC2018-01). The pigs had ad libitum access to food and water and were raised under suitable circumstances.

### 4.2. Animals, Housing, and Experimental Design

Fourteen crossbred growing barrows (Duroc × Landrace × Large White) were used in this experiment. Pigs were surgically fitted with a medical polyethylene cannula (inside diameter 2 mm and outside diameter 3 mm) via an internal jugular vein for the infusion of SB. The experimental period was 4 d after a one-week recovery. After the recovery period, pigs (barrows (BW) = 23.70 ± 1.29 kg) were then randomly allocated into the SB infusion (SB) group and control (Con) group with seven replicates in each group and one pig per pen. At 9:00 of each day during the experimental period, pigs in the SB group were infused with 10 mL of SB (200 mmol/L, PH 7.4, 37 °C) via internal jugular vein, while the Con group was treated with the same dose of physiological saline. The dosage was chosen according to previous studies in pigs including butyrate through parenteral nutrition [24] or through oral administration [6]. During the whole experimental period and recovery period, all pigs were housed into 1.0 × 1.2 m individual pens and fed with commercial feeds with metabolizable energy of 3.19 Mcal/kg and crude protein of 16.8% in as-fed basis. The pigs had ad libitum access to food and water throughout the experimental period.

### 4.3. Sampling

Piglets were fasted overnight on day 4, blood samples of each pig were collected from the precaval vein at 0 (before infusion), 30, 60, and 120 min after infusion, respectively. After sampling, blood samples were centrifuged for 10 min at room temperature, the supernatant samples (serum) were collected and stored at −80 °C for the biochemical indexes and metabolites analysis.

### 4.4. Serum Biochemical Analysis

The concentrations of HDL-C, LDL-C, cholesterol, glucose, and triglyceride in the serum of piglets were determined by an Olympus AU400 Biochemical Analyzer (Tokyo, Japan) following the instrument’s specifications.

### 4.5. Gas Chromatograph-Mass Spectrometry Based Metabolites Profiling

#### 4.5.1. Serum Samples Preparation and GC-MS Analysis

The serum samples were pretreated, extracted, and derivatized as previously reported [26]. Briefly, 100 μL of each sample was mixed with 400 μL methanol containing 2-chlorobenzene alanine (0.2 mg/mL) and heptadecanoic acid (0.2 mg/mL), and vortex-mixed for 1 min. Next, the samples were centrifuged at 12,000 rpm (4 °C) for 10 min, and all serum was dried in a Vacuum centrifugal concentrator (LNG-T83, Huamei, Taicang, China). The dried analytes were mixed with a 60 μL aliquot of methoxyamine pyridine solution (15 mg/mL), then vortex-mixed for 30 s and maintained for 2 h at 37 °C for methoxidation. Then, 60 μL BSTFA reagent was added (containing 1% trimethylchlorosilane), maintained for 90 min at 37 °C, and then the samples were centrifuged at 12,000 rpm (4 °C) for 10 min, and the supernatants were added to the test bottles.

The derivatized samples (1.0 μL) were injected into an Agilent 7890A GC system equipped with a fused-silica capillary column (30 m × 0.25 mm i.d.) and chemically bonded with 0.25 μm DB-5 stationary phase (J&W Scientific, Folsom, CA, USA) by an Agilent 7683 Series autosampler (Agilent Technologies, Atlanta, GA, USA). The inlet temperature was set at 280 °C. Helium was used as the carrier gas with a constant flow rate of 1.0 mL/min through the column. The column temperature was initially maintained at 70 °C for 2 min and then increased at a rate of 10 °C/min from 70 °C to 300 °C, where it was held for 5 min. The column effluent was introduced into the ion source (maintained at 250 °C) of a Pegasus III mass spectrometer (Leco, St. Joseph, MI, USA) through a transfer line, with the temperature set at 250 °C. Mass fragmentation was generated with an electron beam at 70 eV with a current of 3.0 mA. Mass spectra were acquired from *m/z* 35 to 780 at a rate of 30 spectra/s. Following a solvent delay of 170 s, the detector voltage was set to −1650 V.

#### 4.5.2. Data Acquisition and Processing

The retention index of each peak was calculated by comparing the retention time of the peak with that of the alkane series C8 to C40. The compounds were identified by the comparison of the mass spectrum and retention indices of all the detected compounds with their reference standards and database in the National Institute of Standards and Technology Library 2.0 (2008) and the NEW Wiley 9 mass spectra library database. The relative quantitative peak areas of each detected peak were normalized to [13C2]-myristic acid, and the data were arranged on a two-dimensional matrix consisting of arbitrary sample names (observations) and peak area (variables). The original data were transformed into CDF format (NetCDF) using the Agilent GC/MS 5975 Data Analysis software and processed using the XCMS software (v.1.36.0; www.bioconductor.org). The processes of nonlinear retention time alignment, baseline filtration, peak identification, matching, and integration were included. Metabolites analysis and preprocessing were accomplished by SmartNuclide (Suzhou, China).

#### 4.5.3. Univariate, Multivariate, and Pathway Analysis

All statistical analyses of metabolites were done based on different function modules of a powerful tool, which is available online (https://www.metaboanalyst.ca) [45]. Peaks are aligned across all samples based on their mass tolerance and retention time tolerance with the default value of 0.25 (*m/z*) and 5 (s), respectively. Features with >50% missing values were removed and the missing values of the remained features were replaced by a very small value (half of the minimum positive value found in the data set). No missing values were detected with the criterion. The data then underwent logarithmic transformation and normalization of auto-scaling. Fold change analysis and *t*-test were conducted to determine the Fold Change and statistical significance of each metabolite from the serum samples of piglets intravenously infused with SB compared with Con and were further used to select differential metabolites. PCA and PLS-DA were employed to picture the overall difference between the SB and the Con groups and to explore the differential metabolites. The metabolites with variable importance projection (VIP) values above 1.0, FDR (false discovery rate) < 0.1 and Fold Change > 1.5 or < 0.66 were selected as differential metabolites. The differential metabolites were used to execute an enrichment analysis to explore the main pathway changed by the infusion of SB. Time-series + one experimental factor module was used to determine the time-dependent effect, SB infusion effect, and their interaction effect. Venn diagrams were generated by an online tool named Bioinformatics & Evolutionary Genomics (http://bioinformatics.psb.ugent.be/webtools/Venn/).

### 4.6. Data Analysis

Difference in serum biochemical parameters and the relative concentrations of differential metabolites were analyzed by two-way ANOVA using PROC MIXED with time and SB treatment as the main factors in SAS, and an independent sample *t*-test (two-tailed test) was used to compare the data of differential features between two groups at each timepoint (release 8.4; SAS Institute Inc., Cary, NC, USA). Differences with a *p*-value below 0.05 were considered significant.

## 5. Conclusion

In conclusion, intravenous infusion of SB has profound effects on the body metabolism mainly including amino acid metabolism, lipid-related metabolism, and the TCA cycle. Parenteral SB may induce more amino acids to synthesize proteins and affect fat metabolism through increasing fat mobilization and decreasing the biosynthesis of stearates.

## Figures and Tables

**Figure 1 metabolites-10-00020-f001:**
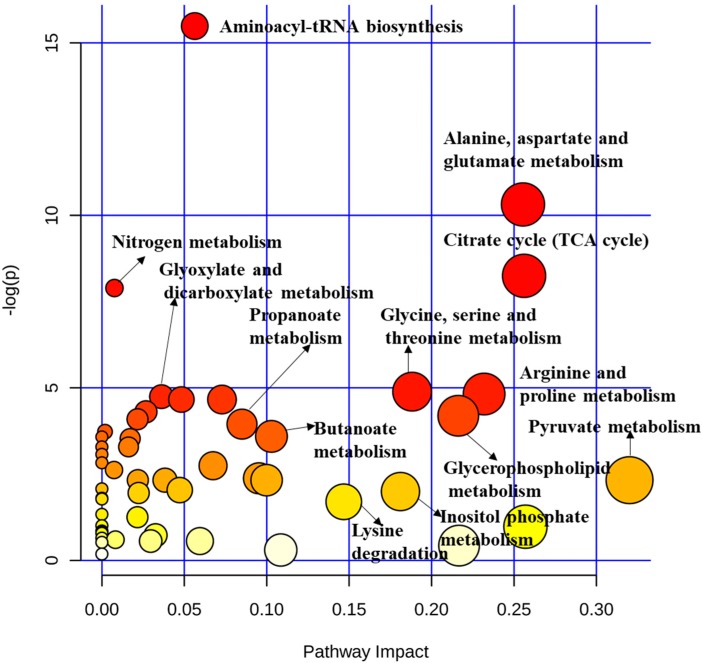
Significantly changed pathways of serum metabolites in pigs affected by the infusion of sodium butyrate (SB) from a whole period perspective. Here, the x-axis represents the pathway impact and the y-axis represents the pathway enrichment. Each node marks a pathway, with larger sizes and darker colors representing higher pathway enrichment and higher pathway impact values.

**Figure 2 metabolites-10-00020-f002:**
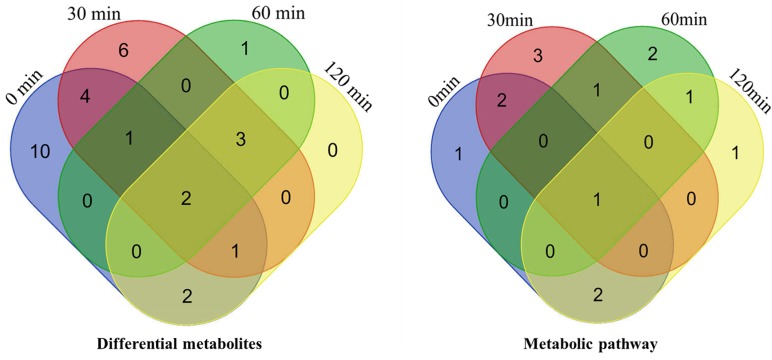
Venn diagrams of differential metabolites and enriched metabolic pathways at the timepoints of 0, 30, 60, and 120 min.

**Figure 3 metabolites-10-00020-f003:**
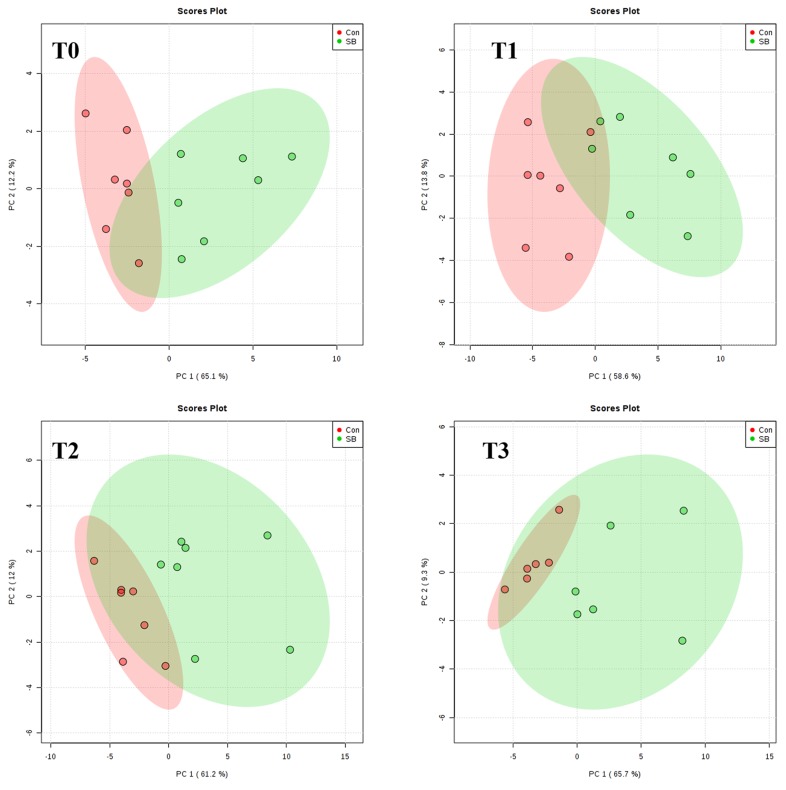
Principal component analysis (PCA) score plot of metabolites of pigs in the control (Con) and sodium butyrate (SB) groups at **T0** (0 min, *n* = 7), **T1** (30 min, *n* = 7), **T2** (60 min, *n* = 7), and **T3** (120 min, *n* = 6) after intravenous infusion.

**Figure 4 metabolites-10-00020-f004:**
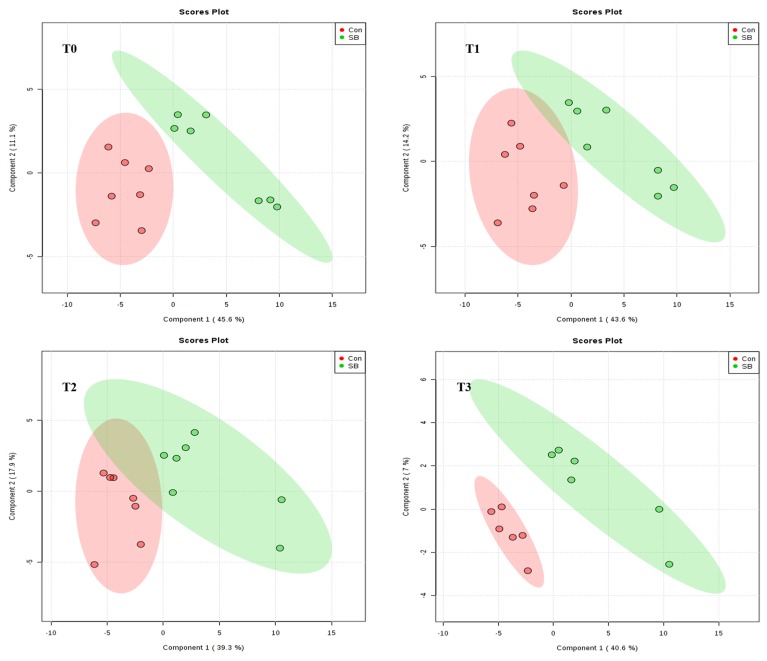
Partial least squares discriminant analysis (PLS-DA) score plot of metabolites of pigs in the control (Con) and sodium butyrate (SB) groups at **T0** (0 min, *n* = 7), **T1** (30 min, *n* = 7), **T2** (60 min, *n* = 7), and **T3** (120 min, *n* = 6) after intravenous infusion. Component 1 = the first principal; Component 2 = the second principal. The explained variances of the first two components are shown in brackets, respectively. The ellipse represents the 95% confidence interval of each group.

**Figure 5 metabolites-10-00020-f005:**
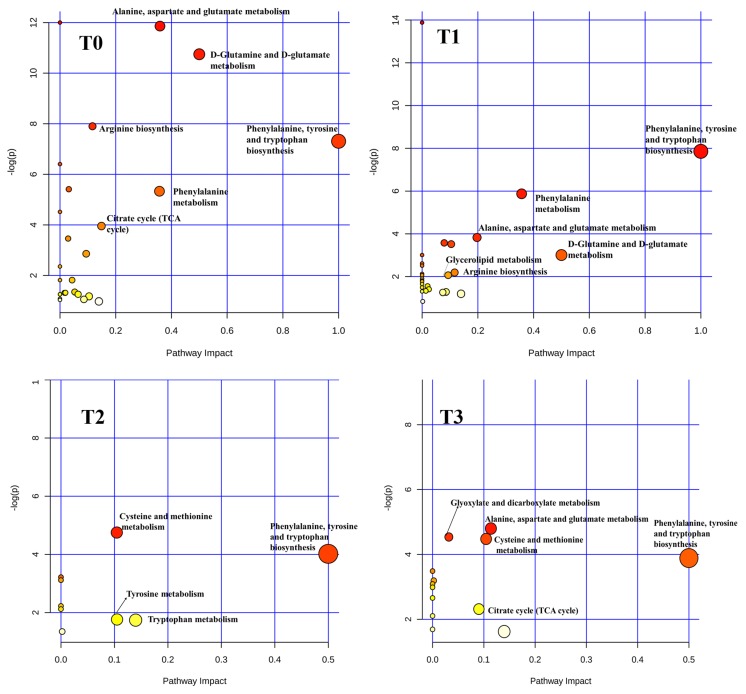
Significantly changed pathways of pigs in the sodium butyrate (SB) group compared with the control (Con) group at **T0** (0 min, *n* = 7), **T1** (30 min, *n* = 7), **T2** (60 min, *n* = 7), and **T3** (120 min, *n* = 6) after intravenous infusion. Here, the *x*-axis marks the pathway impact and the *y*-axis represents the pathway enrichment. Each node marks a pathway, with larger sizes and darker colors representing higher pathway enrichment and higher pathway impact values.

**Figure 6 metabolites-10-00020-f006:**
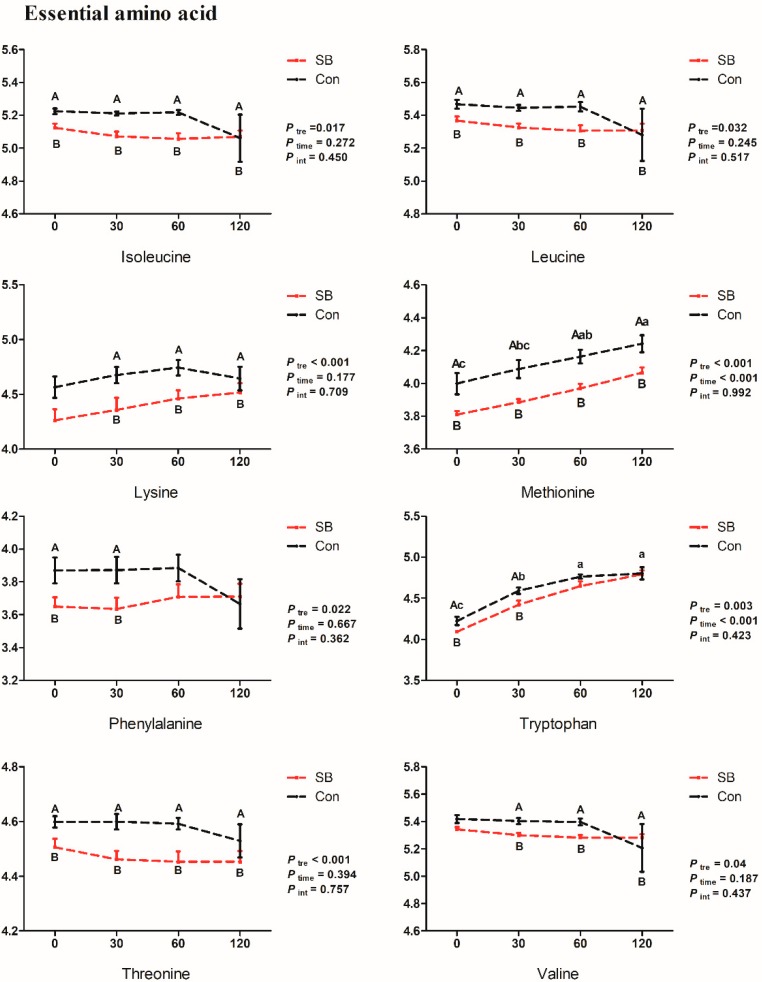
Dynamic changes of the relative concentrations of essential amino acids. X-axis presents the relative concentrates of corresponding metabolites with Log processing of the peak area. Differences were analyzed by two-way ANOVA using PROC MIXED with time and SB treatment as the main factors followed by post-hoc analysis in SAS. Ptre, Ptime, Pint represents the significances of SB treatment, time, and interaction effects, respectively. Superscript with A or B means significant difference between control group (Con) and sodium butyrate group (SB). Superscript with a, b, and c means significant difference over time.

**Figure 7 metabolites-10-00020-f007:**
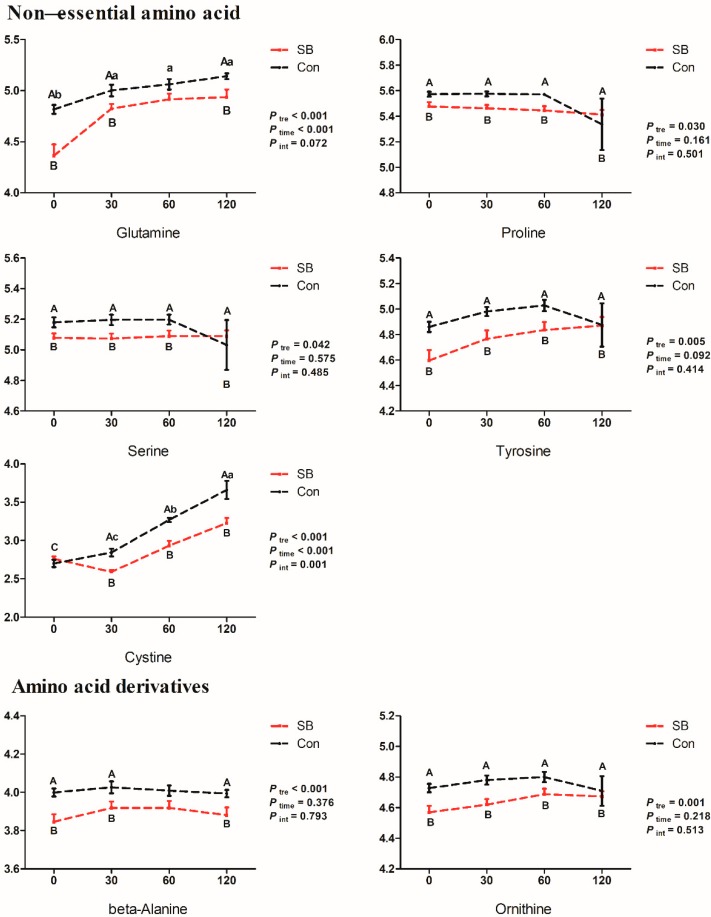
Dynamic changes of the relative concentrations of non-essential amino acids and derivatives. X-axis presents the relative concentrates of corresponding metabolites with Log processing of the peak area. Differences were analyzed by two-way ANOVA using PROC MIXED with time and SB treatment as the main factors followed by post-hoc analysis in SAS. Ptre, Ptime, Pint represents the significances of SB treatment, time, and interaction effects, respectively. Superscript with A or B means significant difference between control group (Con) and sodium butyrate group (SB). Superscript with a, b, and c means significant difference over time.

**Figure 8 metabolites-10-00020-f008:**
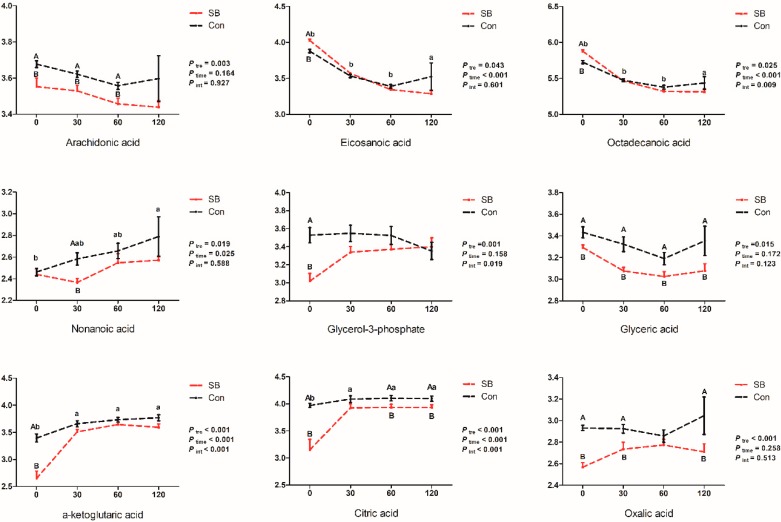
Dynamic changes of the relative concentration of fatty acids and organic acids. X-axis presents the relative concentrations of corresponding metabolites with Log processing of the peak area. Differences were analyzed by two-way ANOVA using PROC MIXED with time and SB treatment as the main factors followed by post-hoc analysis in SAS. Ptre, Ptime, Pint represents the significances of SB treatment, time, and interaction effects, respectively. Superscript with A or B means significant difference between control group (Con) and sodium butyrate group (SB). Superscript with a, b, and c means significant difference over time.

**Table 1 metabolites-10-00020-t001:** Serum metabolite concentrations of pigs in the control (Con) and sodium butyrate (SB) groups at 0, 30, 60, 120 min after intravenous infusion (*n* = 7) ^1^.

Item	Con				SB				SEM	*p* Value		
	0	30	60	120	0	30	60	120		Time	SB	Interaction
**Total protein**	49.83	53.17	50.13	50.79	54.43	51.5	51.49	53.03	2.14	0.787	0.161	0.293
**Triglyceride**	0.37	0.36	0.34	0.40	0.43	0.34	0.4	0.37	0.05	0.608	0.588	0.573
**Cholesterol**	1.94	1.93	1.91	1.96	2.15	2.01	2.04	2.09	0.19	0.930	0.176	0.973
**HDL-C**	0.90	0.94	0.93	0.95	1.13	1.05	1.02	1.06	0.08	0.939	0.005	0.699
**LDL-C**	1.14	1.19	1.14	1.17	1.22	1.16	1.13	1.19	0.13	0.969	0.972	0.927
**Glucose**	4.61	4.69	4.57	4.30	5.05	4.91	4.86	4.93	0.42	0.923	0.101	0.928

^1^ HDL-C = high-density lipoprotein-cholesterol; LDL-C = low-density lipoprotein-cholesterol; SEM = Standard Error of Mean.

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
