# Peer review of "Dynamic Changes in Serum Metabolomic Profiles of Growing Pigs Induced by Intravenous Infusion of Sodium Butyrate"

_metabolites, 2020, doi:10.3390/metabo10010020_

Round 1
Reviewer 1 Report
The manuscript is generally well written. However, there are several things to be addressed, mostly for clarity.
The abstract needs to be shorten. The authors may shorten the description of the experimental design and may remove/shorten some results (such as, PLS-DA) The supplement material is not available, so it was impossible to confirm what the authors claim in the main text. The citations for figures and tables are missing with ‘Error! Reference source not found,’ which also made it difficult to follow the manuscript. Report the PCA analysis results as well. In Figures 4 and 5, did the authors adjust the p-values for multiple comparisons? If not, please do it. In addition, it would be better to put the raw p-values instead of asterisks. Please provide the descriptions for peak alignment and for how missing data were handled. Please use the FDR instead of p-value unless this is a pilot or a proof-of-concept study.
Author Response
Reviewer 1#:
1. The abstract needs to be shorten. The authors may shorten the description of the experimental design and may remove/shorten some results (such as, PLS-DA)
Response: We have simplified the abstract according to your suggestion: Firstly, the background information was deleted (Line 20-22). Secondly, the specific detection method and the results of PLS-DA were removed (Line 30-34); Finally, using “amino acid metabolism” replaced the specific amino acid metabolisms (Line 35-38).
2. The supplement material is not available, so it was impossible to confirm what the authors claim in the main text.
Response: Sorry that you didn't get our supplementary materials. The other reviewer also mentioned that. But in fact, we did prepare it. Anyway, I will check it carefully when uploading the files this time.
3. The citations for figures and tables are missing with ‘Error! Reference source not found,’ which also made it difficult to follow the manuscript.
Response: We must apologize for the problems of “Error! Reference source not found” because of using cross-references when presented the diagrams and tables and not removing the hyperlink before uploading the manuscript. To solve this problem, we will remove all the concerning hyperlinks before uploading the revised manuscript.
4. Report the PCA analysis results as well.
Response: We supplemented the PCA results in Figure 2 (Line 111-120) .
5. In Figures 4 and 5, did the authors adjust the p-values for multiple comparisons? If not, please do it. In addition, it would be better to put the raw p-values instead of asterisks.
Response: According to your suggestion, we regraphed Figure 5 (L178-L182) and Figure 6 (L193-L197).
6.Please provide the descriptions for peak alignment and for how missing data were handled.
Response: We use MetaboAnalyst 4.0 to analyze the metabolomics data. Peaks are aligned across all samples based on their mass tolerance and retention time tolerance with the default value of 0.25 (m/z) and 5 (seconds) respectively. Concerning the handling of missing data, by default, features with > 50% missing values were removed and the missing values of the remained features will be replaced by a very small value (half of the minimum positive value found in the data set) (L319-L323).
7.Please use the FDR instead of p-value unless this is a pilot or a proof-of-concept study.
Response: We supplentmented the FDR in the supplemental materials to replace P-value (L321).
Reviewer 2 Report
This study aims to promote the beneficial effect of sodium butyrate (SB) by examining its impact on the major metabolism of lipid, amino acid and carbohydrate (via TCA) via untargeted metabolomics and bioinformatics approaches. Given the interest of its benefit in well-being, there is interest in this field of research, and I agree with the authors that the impact on host’s metabolism is limited and needs to be explored. The authors showed that amino acid, fat and TCA metabolism were affected by parenteral SB. However, its relevance for health benefit to the host is not described which I think will greatly improve the significance of this study.
Some suggestions to improve the manuscript:
In general, the writing style will need to be improved (for example: in section 3.2, “HDL-C is long considered to be a good kind of cholesterol” needs to be re-written into scientific style.). The referencing index in the main text should use numeric numbers instead of roman numeral that is more commonly accepted. In the introduction, the author mentioned that only a small portion of SB going into the bloodstream via ingestion. Hence, please provide the rationale for the relevance of using 200mmol/L and using infusion administration rather than oral to study the effect of SB. I am surprised to see that the %CV of the QC is really low at <1% for mass spec application. However, there is no supplement material (as stated in the text) provided to justify if this is true. Hence, I can’t comment further on this. What do you mean by “whole period perspective” in Fig 1? Does this mean that the values obtained were calculated by averaging or differences between T3 and T0? On page 9 of 14, “Moreover, the relative concentrations of all ……….. there is no difference in growth performance (data not shown). This sentence is confusing and contradicting where you describe the detected amino acids were decreased by SB but speculate that SB increased the quantality of amino acid in serum. Please clarify. In section 3.2, “Specifically, our result showed that parenteral SB had a chronic effect on increasing ……..indicated that adipokinetic action was increased and fat synthesis was decreased by intravenous infusion of SB.” I disagree that your result has shown anything that indicates “chronic” effect by analyzing up to 120 min. To me, the study is an acute response. Also, do you have any data on visceral fat content or % fat to bodyweight of the pig to demonstrate that the SB impact on the fat metabolism chronically? With the result that SB increases fat catabolism and decreases protein and TCA, how do you know if this effect is not a partial effect of the overnight fasting prior to the treatment?Author Response
1. This study aims to promote the beneficial effect of sodium butyrate (SB) by examining its impact on the major metabolism of lipid, amino acid and carbohydrate (via TCA) via untargeted metabolomics and bioinformatics approaches. Given the interest of its benefit in well-being, there is interest in this field of research, and I agree with the authors that the impact on host’s metabolism is limited and needs to be explored. The authors showed that amino acid, fat and TCA metabolism were affected by parenteral SB. However, its relevance for health benefit to the host is not described which I think will greatly improve the significance of this study.
Response: It’s really a good suggestion, however, since it’s a short-term study focused on the dynamic changes of serum metabolites and related metabolism pathway, all pigs kept healthy during the experiment, it’s hard to include more indicators. However, we will keep your suggestions in mind to improve our subsequent experiment design.
2. In general, the writing style will need to be improved (for example: in section 3.2, “HDL-C is long considered to be a good kind of cholesterol” needs to be re-written into scientific style.).
Response: Thank you for your suggestions to improve our writing style. We will make changes to the corresponding parts of the manuscript.
3. The referencing index in the main text should use numeric numbers instead of roman numeral that is more commonly accepted.
Response: Sorry that the referencing index you saw in the main text were roman numeral. Actually, we did use the numeric numbers when we writing the manuscript. The reason may be that my references were inserted as endnotes in Microsoft word, so when opened in your document it will automatically be displayed according to your endnote Settings. To solve this problem, after revising the manuscript, we will change it into a plain text.
4. In the introduction, the author mentioned that only a small portion of SB going into the bloodstream via ingestion. Hence, please provide the rationale for the relevance of using 200mmol/L and using infusion administration rather than oral to study the effect of SB.
Response: In the section of Materiala and Methods, we added the discription on the rationale for the relevance of using 200mmol/L SB. In Introduction section, we gave the rationale for using infusion administration rather than oral to study the effect of SB.
5. I am surprised to see that the %CV of the QC is really low at <1% for mass spec application. However, there is no supplement material (as stated in the text) provided to justify if this is true. Hence, I can’t comment further on this.
Response: The standard deviations of internal standards in Quality Control were calculated using the retention time which will be attached to the supplemental materials. In addition, we will double check before submitting to make sure you can see our supplementary materials.
6. What do you mean by “whole period perspective” in Fig 1? Does this mean that the values obtained were calculated by averaging or differences between T3 and T0?
Response: The concept of “whole period perspective” in our manuscript means the period from T0 to T3, using which we want to find out the features that were influenced by parenteral SB over time, and the metabolism pathway they further participate in.
7. On page 9 of 14, “Moreover, the relative concentrations of all ……….. there is no difference in growth performance (data not shown). This sentence is confusing and contradicting where you describe the detected amino acids were decreased by SB but speculate that SB increased the quantality of amino acid in serum. Please clarify.
Response: The fact that SB can improve the production performance of animals was well-demonstated and widely accepted. But given it was a short-period trial of only 4 days, it is possible that the beneficial effects of SB have not yet translated into significance improvements in production performance. On the other hand, the concentrations of serum amino acid of SB group were decreased with the same intake of amino acids (feed intake). So we speculated that more amino acids were used in the biosynthesis of proteins. Of course, more studies are needed to confirm this deduction.
8. In section 3.2, “Specifically, our result showed that parenteral SB had a chronic effect on increasing ……..indicated that adipokinetic action was increased and fat synthesis was decreased by intravenous infusion of SB.” I disagree that your result has shown anything that indicates “chronic” effect by analyzing up to 120 min. To me, the study is an acute response.
Response: We initially use this word to distinguish the results of 0 min which timepoint was actually 3 days after the first infusion. But I agree with your point that the word “chronic” was really improper as we didn’t got any data on visceral fat content and other fat accumulation index because of the short experient period.
9. Also, do you have any data on visceral fat content or % fat to bodyweight of the pig to demonstrate that the SB impact on the fat metabolism chronically? With the result that SB increases fat catabolism and decreases protein and TCA, how do you know if this effect is not a partial effect of the overnight fasting prior to the treatment?
Response: Firstly, the results were obtained through comparing with the Con group. Secondly, the pigs had free access to feed and water during the experiment.
Other changes:
Removed all the hyperlinks in the maintext mainly concerning endnotes and cross-reference of tables and figures to ensure all the information are available. Supplemented PCA score plot in Figure 2, so we changed the figure number accordingly. Rephrase some controversial sentences: L43 delete the word “a chronic”; L212-216 Add some necessary premises to make it easier to understand. Delete some nonsense sentences to increase the readability of the article: L22, delete “therefore”; L25, delete “were used in this study. Pigs”; L29-20, delete “at 0, 30, 60 and 120 mins after intravenous infusion” ; L31, delete “(GC-MS).”; L39, delete “detected in this experiment”; L43, delete “while no difference were detected at 30, 60, and 120 mins”; L69, delete “in neonatal piglet”. L204, delete “For a long time”; L225, delete “is long considered to be a good kind of cholesterol. It”. 5. Add some words or sentences to make the expression more clear. L34, add “of metabolomic profiles” ; L70-71, add “inanimal models with intestine injury”; L72, add “in a healthy growing pigmodel”; L218-219, add “while the intake of amino acids (feed intake) was not affected. Thus”; L221-222, add “existed” and “mainly because of the short experiment period”.Round 2
Reviewer 1 Report
Thanks for the responses. There are several things to further consider.
P3, L98: Remove ‘obviously’ It will be good for readers if the authors generate two Venn diagrams to show the consensus among four time points (T0 to T3). One is for the differential metabolites and the other for the pathways detected. 5 and 6: The p-values need to be corrected for multiple comparisons. P11, L293-L295: I believe the Metaboanalyst does not perform the peak alignment anymore. The current statement is not enough to reproduce and I wonder how the authors aligned peaks. Please provide a detailed description for the peak alignment as well as the name of software packages if used. P11, L293-L295: How many peaks were aligned after peak alignment? P11, L295-L297: “the missing values of the remained … will be replaced by a very small value (half of … set)” è “the missing values of the remained features was replaced by a half of the minimum positive value found in the data set.” P11, L295-L297: How many peaks were removed after filtering out with >50% missing values? P11, L303: The fold change of 1.2 (or 0.83) is too small. This should be at least 1.5 (or 1/1.5). Please redo the analysis with the fold change of 1.5 (or 1/1.5). P11, L308: Which analysis did you use ANOVA? If it wasn’t, please remove it. Section 2.3 and Supplementary Tables S4 to S7: It appears that all differential metabolites are downregulated at all time points except at time =0. At time =0, there are only three metabolites upregulated. So, I wonder whether these are expected. Please discuss it.
Author Response
P3, L98: Remove ‘obviously’ It will be good for readers if the authors generate two Venn diagrams to show the consensus among four time points (T0 to T3). One is for the differential metabolites and the other for the pathways detected.
Response: Thank you for your suggestion. Firstly, We supplement two Venn diagrams both for the differential metabolites and the metabolic pathways (Firgure 2). Secondly, we supplement details of venn diagrams (L96-L99, L328-329). Thirdly, we deleted the word “obviously” (L105).
5 and 6: The p-values need to be corrected for multiple comparisons.
Response: In deed, the p-value noted were obtained from the results of t-test of each timepoint, as we didn’t execute multiple comparisons of the four timepoints here.
P11, L293-L295: I believe the Metaboanalyst does not perform the peak alignment anymore. The current statement is not enough to reproduce and I wonder how the authors aligned peaks. Please provide a detailed description for the peak alignment as well as the name of software packages if used.
Response: Sorry that the written information was not adequate. As the pretreatment process were mainly finished by commercial company, we further consulted them for detail information. We will list the details below and supplement them in our manuscript (L291-L294). Specifically, the original data were transformed into CDF format (NetCDF) using Agilent GC/MS 5975 Data Analysis software and processed using XCMS software (www.bioconductor.org). The processes of nonlinear retention time alignment, baseline filtration, peak identification, matching and integration were included.
P11, L293-L295: How many peaks were aligned after peak alignment?
Response: As reported by SmartNuclide, there are 232 peaks were obtained.
P11, L295-L297: “the missing values of the remained … will be replaced by a very small value (half of … set)” è “the missing values of the remained features was replaced by a half of the minimum positive value found in the data set.” P11, L295-L297: How many peaks were removed after filtering out with >50% missing values?
Response: According to the above standard, a total of 0 (0%) missing values were detected.
P11, L309: The fold change of 1.2 (or 0.83) is too small. This should be at least 1.5 (or 1/1.5). Please redo the analysis with the fold change of 1.5 (or 1/1.5).
Response: We really appreciate your preciseness on research. Thus we take your suggestion and redo the analysis with the fold change of 1.5 (or 0.66). The results of differential metabolites showed in Supplemental Materials Table S4 (0 min), Table S5 (30 min), Table S6 (60 min), and Table S7 (120 min) respectively. Accordingly, we revised the results of pathway analysis in the manuscript (L119-120; L130-133; L145-149; L151-154) and replaced Figure 5 (L129).
P11, L308: Which analysis did you use ANOVA? If it wasn’t, please remove it.
Response: Accroding to your suggestion, we revised these sentence as “Difference in serum biochemical parameters were analyzed by two-way ANOVA using PROC MIXED with time and SB treatment as the main factors in SAS,”.
Section 2.3 and Supplementary Tables S4 to S7: It appears that all differential metabolites are downregulated at all time points except at time =0. At time =0, there are only three metabolites upregulated. So, I wonder whether these are expected. Please discuss it.
Response: As a short-term study in just 120 min, It’s possible that parenteral SB has an acute effects on decreasing the relative concentration of these metabolites. What’s more, at 0 min (24h after the last infusion) the increasing concentration of some metabolites were detected.
Reviewer 2 Report
I'm happy with the correction that the authors had made. One final comment that i have a little concern is the style of writing will still need some tweaking and suggests that the authors with the help from editor to work on it.
Author Response
Thanks for your approval, major changes were made to improve the manuscript in the revised version.
Round 3
Reviewer 1 Report
Thanks for the responses, but there are still several things to be addressed yet.
For Fig. 6 and 7, the p-values should be corrected for the multiple comparisons. In addition, please provide which method is used for multiple comparison correction in the legend of each figure. Please replace ‘NS’ with the raw p-values up to three decimal digits. Please include a statement that the metabolite experiments and preprocessing were done may the company ‘SmartNuclide’ in Section 4.5.2. Please add a statement about the number of peaks aligned in the main text. Please add a statement that no missing values were detected with the criterion (>50% missing values). In Section 4.5.3, the sentence ‘Features with <50% ….” Is duplicated Please add a discussion about my previous comment ‘Section 2.3 and Supplementary Tables S4 to S7: It appears that all differential metabolites are downregulated at all time points except at time =0. At time =0, there are only three metabolites upregulated. So, I wonder whether these are expected’ in the discussion section.Author Response
For Fig. 6 and 7, the p-values should be corrected for the multiple comparisons. In addition, please provide which method is used for multiple comparison correction in the legend of each figure.Response: According to your suggestion, we had reanalyzed the relative concentrations by two-way ANOVA using PROC MIXED with time and SB treatment as the main factors in SAS while replaced Fig. 6 and 7 (L161 and L179). And we add a statement in the legend of the figure.
Please replace ‘NS’ with the raw p-values up to three decimal digits.Response: According to your suggestion, we has replace ‘NS’ in Fig. 6 and 7.
Please include a statement that the metabolite experiments and preprocessing were done may the company ‘SmartNuclide’ in Section 4.5.2.Response: we added the statement that “Metabolites analysis and preprocessing were accomplished by SmartNuclide” in L296.
Please add a statement about the number of peaks aligned in the main text.Response: we added “No missing values were detected with the criterion.” in L305.
Please add a statement that no missing values were detected with the criterion (>50% missing values).Response: we added the statement that “No missing values were detected with the criterion.” in L305.
In Section 4.5.3, the sentence ‘Features with <50% ….” Is duplicatedResponse: we had deleted the duplicated sentence (L299-L301).
Please add a discussion about my previous comment ‘Section 2.3 and Supplementary Tables S4 to S7: It appears that all differential metabolites are downregulated at all time points except at time =0. At time =0, there are only three metabolites upregulated. So, I wonder whether these are expected’ in the discussion section.Response: Thanks for your suggestion, we had add more information in the discussion parts (L205-L212).